# Music audio emotion regression using the fusion of convolutional neural networks and bidirectional long short-term memory models



Yi Qiu, Yu Lin and Yun Lin

School of Information Science and Technology, Xiamen University Tan Kah Kee College, Zhangzhou, China

## ABSTRACT

Music emotion regression (MER) is a vital field that bridges psychology and music information retrieval. Music has the powerful ability to evoke a wide range of human emotions, from joy and sadness to anger and calmness. Understanding how music influences emotional states is essential for grasping its psychological effects on individuals. This research presents an innovative model that combines convolutional neural networks (CNNs) with bidirectional long short-term memory (BiLSTM) networks to analyze and predict the emotional impact of musical audio. The model uses CNNs to detect temporal patterns and BiLSTMs to interpret sequences in both forward and backward directions, enhancing its ability to capture the complex structure of musical data. Additionally, a multi-head attention mechanism is incorporated to improve the model's expressiveness and generalizability, making it especially effective for handling intricate sequential tasks and large datasets. The model's performance was evaluated through sentiment prediction using extensive, publicly available datasets comprising over 9,000 musical excerpts. Results show that the proposed model significantly outperforms existing methods in MER, achieving an R-squared value of 0.845, indicating an excellent fit with the empirical data.

Corresponding author
Yun Lin, yunlin202410@163.com

## INTRODUCTION

Today's youth appear to be more sensitive and fragile compared to previous generations. In recent years, they have experienced a range of psychological issues, including elevated stress levels, anxiety, depression, loneliness, and social media addiction (*Wang et al., 2021*). It is crucial to pay attention to the emotional changes occurring in young people, as this understanding can shed light on their attitudes and emotional tendencies. Music emotion recognition (MER) has gained significant attention in addressing these challenges (*Han et al., 2022*; *Panda, Malheiro & Paiva, 2020*; *Huang et al., 2022*).

MER is a field of study that focuses on using computer algorithms and techniques to analyze and identify emotions conveyed in music. Researchers in this area aim to create systems that can automatically detect and classify the emotional content of musical pieces, determining whether a piece sounds happy, sad, calm, or exciting. MER has several

applications, including music recommendation systems, music therapy, and affective computing (*Gómez-Cañón et al., 2021*; *Koh et al., 2022*). MER plays a crucial role in understanding the complex relationship between music and human emotions. It helps psychologists quantify the emotional experiences that music can evoke, which is essential for studying the neural and psychological mechanisms behind human emotions. By predicting emotional responses to music, researchers gain insights into how various musical elements contribute to emotional arousal and valence. There are a variety of methods and techniques used in MER to analyze and predict the emotional content of music (*Hizlisoy, Yildirim & Tufekci, 2021*; *Chaturvedi et al., 2022*; *Han et al., 2022*).

Traditionally, various machine learning algorithms such as support vector machines (SVM), k-nearest neighbors (KNN), decision trees (DT), random forests (RF), and naïve Bayes (NB) are employed to implement MER systems (*Han et al., 2022*; *Yang, 2021*). In the context of MER, SVM can be trained on a set of musical features to identify patterns associated with different emotional states. These features can include audio characteristics like tempo, melody, harmony, rhythm, and timbre, as well as semantic features derived from lyrics in vocal music. Once trained, the SVM model can predict the emotional content of unknown music samples based on these features (*Devendran et al., 2021*; *Kawintiranon, Buatong & Vateekul, 2016*). For example, *Devendran et al. (2021)* investigated various machine learning algorithms to predict the relaxation effects of music for therapeutic purposes. Their study compared methods such as decision trees, random forests, artificial neural networks (ANN), SVM, and a hybrid of SVM and ANN, finding that the hybrid model was the most accurate. This suggests that the SVM-ANN hybrid could serve as a valuable tool in music therapy for relaxation, particularly when working with large datasets or high-dimensional feature spaces.

Deep learning architectures, such as convolutional neural networks (CNNs) and recurrent neural networks (RNNs), are increasingly employed to automatically extract features from raw audio data for MER (*Yu et al., 2017*; *Sams & Zahra, 2023*; *Rajesh & Nalini, 2020*). In the study conducted by *Rajesh & Nalini (2020)*, a novel approach is introduced that utilizes RNN techniques to recognize the emotions conveyed by musical instruments. The findings suggest that deep RNNs deliver superior performance for MER. However, RNNs may face challenges such as the vanishing or exploding gradient problem, particularly when dealing with long sequences. This issue can negatively affect the accuracy of music emotion predictions (*Ribeiro et al., 2020*; *Rehmer & Kroll, 2020*).

Despite their limitations, CNNs and RNNs can be effective in MER when applied correctly and in conjunction with other techniques (*Li et al., 2020*; *Prabhakar et al., 2023*; *Du, Li & Gao, 2020*). For instance, *Li et al. (2020)* propose a multichannel long short-term memory (LSTM) RNN architecture for emotion prediction. This approach effectively learns feature representations and models the temporal dependencies between their activations (*Li et al., 2020*). The experimental results demonstrate an improvement over existing methods.

Motivated by previous studies, the objective of this article is to integrate various techniques, including feature extraction, machine learning, and deep learning, to enhance the accuracy of emotion prediction models. This study proposes a framework based on a

convolutional neural network-bidirectional long short-term memory (CNN-BiLSTM) architecture. The CNN component includes a multi-scale processing module, which establishes a sparse structure in the model's architecture rather than a dense one. Additionally, residual blocks are introduced to increase the model's depth and improve its learning capability. The convolutional neural layer effectively extracts essential information, while the BiLSTM neural network is capable of processing bidirectional time series data in the temporal dimension, leading to more comprehensive and accurate predictions (*Jang et al., 2020*). CNNs are particularly adept at capturing local features, whereas BiLSTM networks integrate both forward and backward time series data, allowing for improved analysis of multivariate time series in the context of music emotions. Furthermore, the incorporation of a multi-head self-attention mechanism enhances the model's expressive power and generalization performance, making it well-suited for tackling complex sequence tasks and handling large datasets (*Yang et al., 2023*).

Our work contributes to the field of music therapy by providing a precise tool that predicts the emotional impact of music. By accurately measuring how different pieces of music evoke specific emotions, our model helps therapists customize music interventions to better meet the unique needs of individual patients. This is particularly beneficial for treating psychological conditions such as anxiety, depression, and post-traumatic stress disorder (PTSD), where music therapy has shown effectiveness. Our approach not only enhances the understanding of the complex relationship between music and emotions but also supports the development of practical technologies aimed at improving mental health and overall societal well-being.

The main contributions of our article are as follows:

- We have developed an integrated CNN-BiLSTM model specifically designed for the regression task of predicting emotional responses to music audio. This model serves as a valuable tool in music therapy, providing targeted treatments for psychological conditions such as anxiety, depression, and PTSD by accurately quantifying and predicting the emotional impact of music. Our approach not only deepens the understanding of the complex relationship between music and emotional states but also drives the development of practical technologies aimed at enhancing mental health and promoting societal well-being.

- The combination of CNN and BiLSTM networks provides several advantages. Firstly, CNNs are effective at extracting hierarchical features from input data, such as audio spectrograms in MER, capturing both local and global patterns. In contrast, BiLSTMs are particularly skilled at modeling temporal dependencies. By integrating both architectures, the model can leverage the strengths of each for enhanced feature extraction. Additionally, the bidirectional nature of BiLSTMs allows the model to gather information from both directions of the input sequence, facilitating a more comprehensive analysis of the temporal relationships present in music data.

- The combined CNN and BiLSTM system has been improved with the addition of a multi-head attention mechanism. This enhancement allows the model to focus on different parts of the input sequence simultaneously, enabling it to learn more diverse

and informative representations of the music data. As a result, the model gains a deeper understanding of the emotional content present in the music. Additionally, integrating multi-head attention into MER models can lead to more robust, accurate, and expressive representations of music data. This, in turn, enhances the performance of emotion prediction systems in analyzing and understanding the emotional aspects of music.

- The combination of CNNs for spatial feature extraction and BiLSTMs for temporal modeling often results in improved accuracy in emotion recognition tasks. Experiments on over 9,000 music datasets indicate an R-squared of 0.845.

- Our proposed CNN-BiLSTM model enables accurate identification and prediction of emotions from music, which can significantly enhance psychological therapy. As music serves as a powerful tool for emotional expression, our model can help therapists select pieces tailored to evoke specific emotional states, such as calming music to reduce anxiety or uplifting music to improve mood in individuals with depression. Furthermore, the model can assist in creating personalized music therapy sessions that align emotional content with therapeutic goals, ultimately boosting patient engagement and improving outcomes.

The article is structured as follows: We first introduce our proposed model in "Methodology", and then we describe the test bed. We evaluate the proposed model according to several evaluation metrics in "Implementation and Experiments". After assessing the performance of the proposed model, we present the summary and discuss future work in "Conclusion".

## METHODOLOGY

The process of music sentiment regression in our model is shown in Fig. 1. First, we use Word2Vec to preprocess the input data, referencing the works of *Di Gennaro, Buonanno & Palmieri (2021)*, *Grohe (2020)*. Word2Vec converts each word in the input sequence into a high-dimensional vector representation that captures semantic and syntactic information about the music audio. Next, we employ CNN layers to extract local features from the input embeddings, capturing patterns and relationships between neighboring words. The output of the CNN layers is then downsampled using MaxPooling layers to reduce feature dimensionality while retaining important information. After this, the outputs are fed into BiLSTM layers, which are capable of capturing long-range dependencies in the input sequence by processing the data in both forward and backward directions. Following the BiLSTM layer, the outputs are processed through a fully connected (FC) layer, integrating the learned features across different parts of the input sequence. This layer combines the features into a comprehensive representation that captures the overall sentiment of the music. Prior to the final prediction, an attention layer is strategically introduced right after the FC layer. This attention mechanism serves a pivotal function in refining the integrated features by applying a final weighting process, which effectively highlights the most significant feature dimensions. The attention layer operates based on the principle of focusing on the most informative parts of the data. It achieves this by computing a set of attention weights that are learned during the training

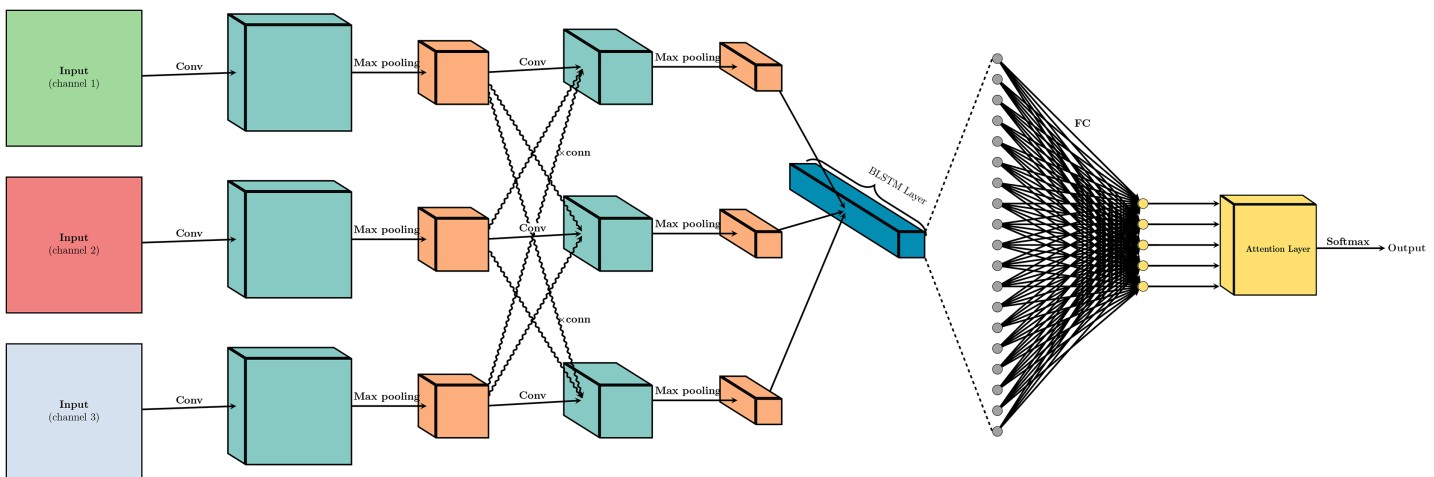

**Figure 1  The system architecture of the proposed model.**

process. These weights are determined by an attention scoring function that assesses the relevance of each feature dimension to the task at hand. The scoring function typically involves a small neural network that takes the output of the FC layer as input and generates a weight for each feature dimension. Finally, a Softmax layer is applied to the output of the attention layer to perform the final regression task. The Softmax function ensures that the output is a probability distribution over the sentiment scores, allowing the model to predict a continuous sentiment value rather than a discrete category. This approach enables the model to provide a nuanced and fine-grained sentiment prediction, reflecting the subtleties in the music's emotional content.

## Feature extraction layer

In order to better capture local emotional characteristics from historical information feature vectors and music emotion feature vectors, we have incorporated CNN and MaxPooling layers into the model. When assessing the emotional content of a song in a continuous dimensional manner, the music emotion feature matrix is first fed into a one-dimensional CNN layer. The input layer is made up of a one-dimensional tensor representing a time series, with the input data dimensions being $N \times T \times F$, where $N$ is the number of samples, $T$ is the number of time steps, and $F$ is the feature dimension at each time step.

In Fig. 2, the input data is processed sequentially through a series of layers designed for feature extraction and dimensionality reduction. The network starts with an input layer, followed by a number of convolutional layers (labeled 'Conv'). These layers use learnable filters to perform convolution operations that capture spatial hierarchies within the input. Batch normalization (BN) layers are interspersed among the convolutional layers to normalize the inputs, accelerate convergence, and mitigate internal covariate shifts during the training process. The network also incorporates rectified linear unit (ReLU) activation functions (labeled 'ReLU') to introduce non-linear transformations, allowing the network to learn complex patterns and representations. Additionally, the architecture features

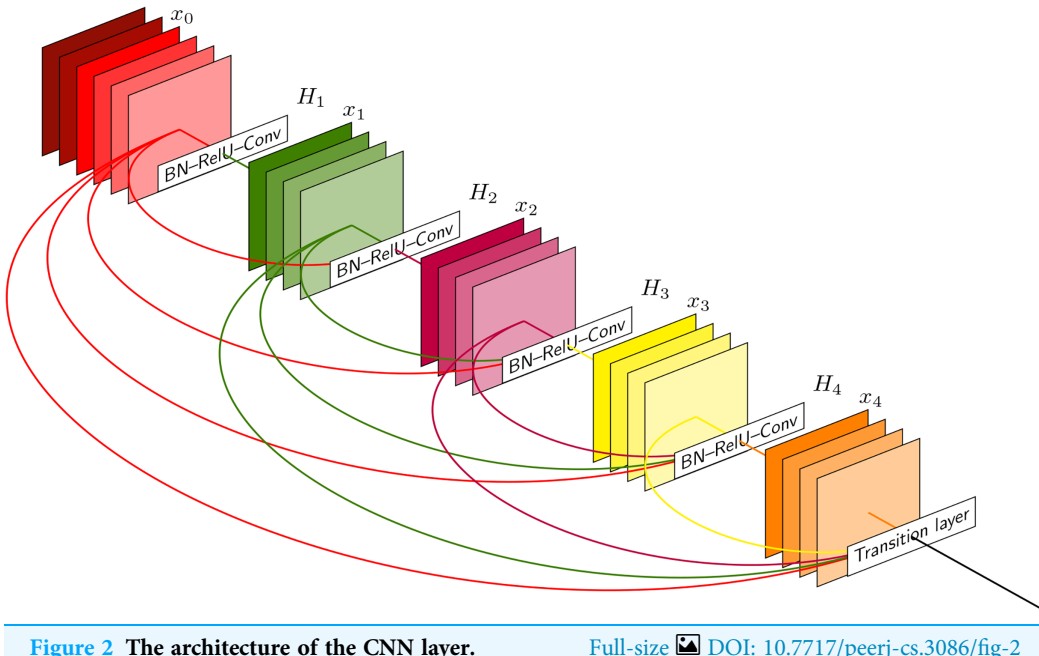

**Figure 2 The architecture of the CNN layer.**

layers where ReLU is combined with batch normalization (labeled 'BN-ReLU'), indicating an improvement in the network's ability to learn from the normalized inputs. The transition layer condenses the spatial dimensions of the feature maps, preparing the data for subsequent layers while preserving essential features. The network progresses through hierarchical layers labeled 'H1' to 'H4', with the numerical suffix indicating increasing depth and complexity of the learned features. This structured approach enables the CNN to progressively abstract and refine the representations of the input data as it moves through the network's layers.

## BiLSTM layer

We use the BiLSTM layer to address the issue of vanishing gradients. AS depicted in Fig. 3, the BiLSTM layer is designed to address the limitations of unidirectional LSTMs by processing data sequences in both forward and backward directions, which is essential for its ability to capture and retain information across various time frames. In the forward layer, the sequence is processed from left to right, starting from $x_1$ to $x_t$, generating forward hidden states $h_t^f$ at each time step. Conversely, the backward layer processes the sequence from right to left, producing backward hidden states $h_t^b$. Each layer employs a gating mechanism that includes input, forget, and output gates to regulate the flow of information, thereby preventing the vanishing gradient problem.

At the core of this mechanism are the gates, comprising structures such as the input gate, forget gate, and output gate.

The input gate determines the amount of new input information that will be stored in the cell state. It uses a sigmoid layer to determine which values to update, allowing certain information to pass through while blocking others, ensuring that only relevant

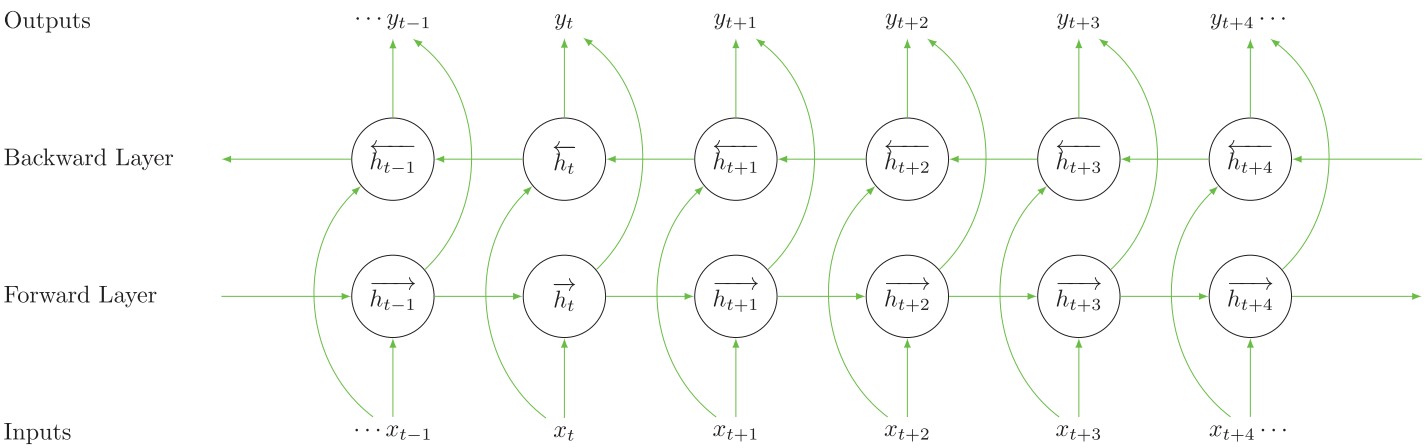

**Figure 3 The architecture of the BiLSTM model.**     

information is retained. Conversely, the forget gate decides which information should be discarded or overlooked from the cell state. It also uses a sigmoid layer to assess the entire cell state and determine which parts of the previous information are no longer necessary and should be forgotten. The cell state itself acts as a form of memory that retains information across different time steps. It is updated by the input gate and forget gate, with new information being added and outdated information being removed, respectively. The output gate determines which information is used to calculate the next hidden state and the output of the LSTM cell. It consists of a sigmoid layer to select the relevant parts of the cell state and a tanh layer to add non-linearity to the output.

In a BiLSTM, this gating process occurs in both forward and backward directions, allowing the model to process information from both past and future contexts. This bidirectional capability makes BiLSTM particularly effective for tasks requiring understanding of context in both directions, such as speech recognition, language translation, and sentiment analysis. The gates work together to enable the model to focus on relevant information, retain essential details, and discard outdated or irrelevant data, all while considering the broader temporal context.

In Eq. (1), the variables $i, f, o, g, c$, and $h$ represent different components of the BiLSTM layer. These components include the input gate, forget gate, output gate, cell gate, cell state, and hidden state for both the forward and backward LSTMs. The input gate $i_t^f$ controls the flow of information to the cell state, while the forget gate $f_t^f$ decides what data to discard from the cell state. The output gate $o_t^f$ determines what data to output based on the cell state. These gates are calculated using a sigmoid activation function based on the previous hidden state $h_t^f$ and the current input $x_t$. Additionally, the gate $g_t^f$ controls what information to output based on the cell state and is calculated using a tanh activation function. The cell state $c_t^f$ is updated by considering the input gate, forget gate, and candidate value. It determines what information to keep in memory and forget. Similarly, the backward LSTM computes the hidden state $h_t^b$ at time step $t$ using Eq. (1). The

equations for the backward LSTM are similar to those of the forward LSTM, but they process the input sequence in reverse order, capturing information from future time steps.

Our proposed BiLSTM architecture differs from traditional LSTM models in that it can process sequences in both forward and backward directions. This dual processing can be mathematically represented as $h_t = [h_t^f; h_t^b]$, where $h_t^f$ refers to the forward hidden states and $h_t^b$ refers to the backward hidden states. By combining these forward and backward hidden states, the BiLSTM can integrate information from both directions, resulting in a more comprehensive representation of the input sequence. This feature is particularly beneficial for tasks such as music emotion analysis, where understanding the emotional content often requires considering both the preceding and following musical elements.

$$
\begin{aligned}
i_t^b &= \sigma(W_i^b \cdot [h_{t+1}^b, x_t] + b_i^b) \\
f_t^b &= \sigma(W_f^b \cdot [h_{t+1}^b, x_t] + b_f^b) \\
o_t^b &= \sigma(W_o^b \cdot [h_{t+1}^b, x_t] + b_o^b) \\
g_t^b &= \tanh(W_g^b \cdot [h_{t+1}^b, x_t] + b_g^b) \\
c_t^b &= f_t^b \odot c_{t+1}^b + i_t^b \odot g_t^b \\
h_t^b &= o_t^b \odot \tanh(c_t^b).
\end{aligned}
\tag{1}
$$

By combining the outputs of the forward and backward LSTMs, a BiLSTM layer can capture dependencies from both past and future contexts, making it practical for tasks requiring a comprehensive understanding of sequential data.

## IMPLEMENTATION AND EXPERIMENTS

Our experiments were conducted on an Ubuntu 16.04.7 LTS system optimized to enhance the performance of our MATLAB 2023a model. To ensure swift processing, we utilized an graphics processing unit (GPU, NVIDIA GeForce RTX 2080 Ti) along with an central processing unit (CPU, RTX 2070). This setting provided the computational power necessary for our model, allowing us to achieve efficient and accurate experimental results.

Our study utilizes the well-respected valence-arousal model as the foundation for analyzing and predicting the emotional dimensions of music. This model is a crucial element in the field of MER and provides a systematic framework for identifying and measuring the range of emotional responses that music can evoke (*Bai et al., 2016*). Widely recognized in psychology and affective science, the valence-arousal model categorizes emotional states along two perpendicular axes: valence and arousal. In therapeutic settings, this model is a valuable tool for clinicians, helping them select musical stimuli that align with specific emotional states, thus aiding in the achievement of therapeutic goals.

To thoroughly assess the accuracy and reliability of our system, we compare its predictions of valence and arousal with established benchmarks and human-annotated data. The foundation of our effectiveness evaluation is based on the DEAM and PMEmo datasets (*Arablouei et al., 2021*; *Zhang et al., 2018*). The DEAM dataset, which is accessible at DEAM Dataset - Emotional Analysis in Music, provides valuable resources for analyzing

emotions in music. It contains 1,802 musical pieces and provides both second-by-second and overall annotations of valence and arousal. PMEmo, a leading dataset in this field, is known for its carefully selected annotators and the inclusion of simultaneous physiological measurements, all drawn from a wide range of musical selections. The PMEmo dataset can be accessed at the PMEmo Repository. However, it is important to note that these datasets may exhibit certain biases that could affect our model's training and testing outcomes. One potential bias is the demographic composition of the annotators, which may skew the emotional labels toward specific cultural interpretations. Additionally, the datasets might not capture the full diversity of musical genres, leading to a model that performs well on the training data but generalizes poorly to new, unseen data. In this study, we did not fully address potential dataset biases, which we recognize as an important area for future work. We plan to incorporate experiments into our upcoming research to assess and mitigate these biases, ensuring the robustness of our model across diverse contexts.

In the data processing phase, we began by loading the DEAM and PMEmo datasets into a Jupyter Notebook, utilizing the panda's library for effective data manipulation. To tackle the common issue of missing data found in real-world datasets, we employed a forward-filling strategy to maintain the sequential integrity of the emotional annotations. Following this, we standardized the feature space using Z-score normalization to mitigate the effects of varying units and scales among different features. These datasets are segmented into training, development, and testing subsets, with an allocation ratio of 80% for training and 10% apiece for development and testing phases.

We optimized our music emotion regression model by carefully selecting hyperparameters and enhancing the training process. During preprocessing, we used a Random Forest algorithm to determine feature importance, filtering out features below a threshold of 0.0015. This, combined with Z-score normalization, ensured our model focused on the most informative features while reducing the impact of varying scales.

For the neural network, we set a learning rate of 0.001, a batch size of 32, and 128 hidden units, balancing complexity and training efficiency. We used the Adam optimizer for its adaptive learning rates and implemented early stopping to avoid overfitting. Additionally, we applied data augmentation techniques, like adding noise to the input data, to improve generalization. These strategies enhanced the robustness and accuracy of our model.

All the code for the project has been deposited in the GitHub repository, which can be accessed and downloaded through the following link: Music-Emotion-Recognition. This repository contains all the code for data preprocessing, neural network construction, and experimental setup, facilitating your research and development in the field of music emotion recognition.

## Factors affecting the model quality

Entropy, as defined by the Noisy-Channel Coding Theorem, is a fundamental concept in information theory (*Hui & Belkin, 2020*; *Attanasio et al., 2022*). It measures information uncertainty and can be thought of as a measure of noise obscuring information. In this experiment, we use entropy to explore the degree of dispersion of a specific indicator. A

smaller entropy value for the indicator indicates a greater degree of dispersion and a more significant impact on evaluation. The process of calculating is as follows:

- We use different algorithms for data standardization processing for high and low indicators. When calculating positive indicators, we use Eqs. (2), and (3) is used for calculating negative indicators.

$$z_{ij} = \frac{x_{ij} - \min(x_{\cdot j})}{\max(x_{\cdot j}) - \min(x_{\cdot j})} \tag{2}$$

$$z_{ij} = \frac{\max(x_{\cdot j}) - x_{ij}}{\max(x_{\cdot j}) - \min(x_{\cdot j})} \tag{3}$$

where $z_{ij}$ is the normalized value of the $j$-th indicator for the $i$-th sample after standardization, $x_{ij}$ is the original value, and $x_{\cdot j}$ represents all values of the $j$-th indicator across samples.

- Calculate the proportion of the $i$-th sample in the $j$-th indicator:

$$w_{ij} = \frac{x_{ij}}{\sum_{i=1}^{n} x_{ij}} \tag{4}$$

It computes the proportion of the $i$-th sample in the $j$-th indicator by dividing the value of the sample by the sum of all values of the $j$-th indicator across samples.

- Calculate the entropy of the $j$-th indicator:

$$E_j = -\sum_{i=1}^{n} w_{ij} \ln(w_{ij}) \tag{5}$$

where ln represents the natural logarithm, $n$ is the number of samples, and $C$ is a constant related to the sample size $n$. Generally, if $C = 1$, then $E_j = 1$.

- Calculate the information utility value of the $j$-th indicator:

$$I_j = 1 - \frac{E_j}{\ln(n)} \tag{6}$$

- Calculate the weights of each indicator:

$$W_j = \frac{I_j}{\sum_{j=1}^{m} I_j} \tag{7}$$

- Calculate the comprehensive score of each sample:

$$S_i = \sum_{j=1}^{m} W_j \cdot x_{ij} \tag{8}$$

We use MAE, MSE, R-Square ($R^2$), MAPE, and RMSE as evaluation metrics. Mean absolute error (MAE) measures the average magnitude of prediction errors, with lower values indicating better accuracy. Mean squared error (MSE) emphasizes larger errors by squaring them, providing a sensitivity to outliers. The coefficient of determination, R-Square ($R^2$), reveals the proportion of variance in the dependent variable that is predictable from the independent variables, with values closer to 1 indicating a better

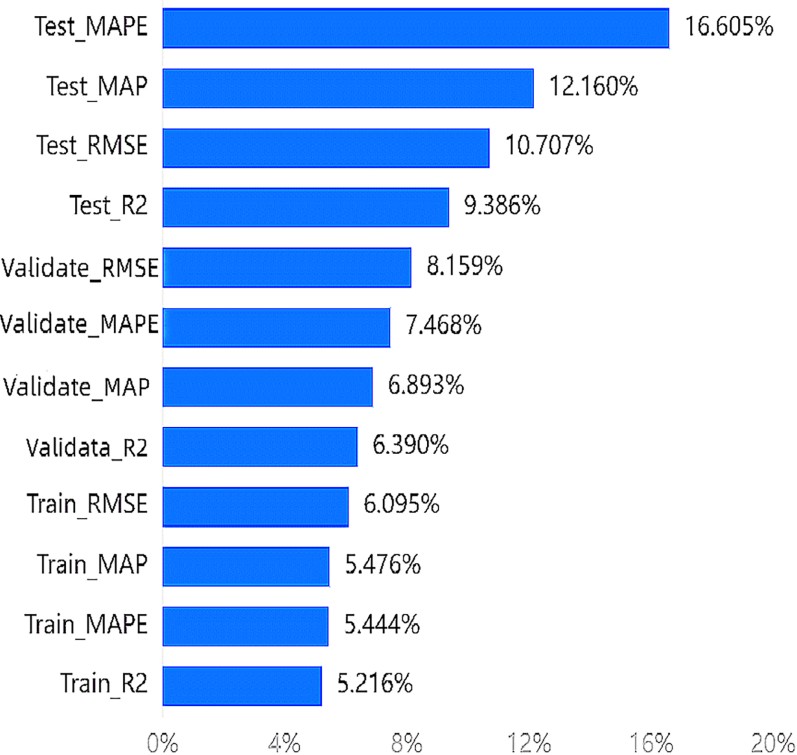

**Figure 4** **Importance ranking of training indicators in the wakefulness dimension.**

model fit. Mean absolute percentage error (MAPE) assesses prediction accuracy in percentage terms, with lower values being more desirable. Lastly, root mean squared error (RMSE) offers an interpretable measure of the magnitude of prediction errors in the same units as the data. For neural networks, a higher $R^2$ and lower MAE, MAPE, and RMSE values are generally preferred, signifying more accurate and reliable predictions.

According to the entropy method, the weight results indicate that the highest weight among the indicators is 16.605% for *Test_MAPE*, and the lowest weight is 5.216% for *Train_R²*. All four indicators of the testing set rank in the top four. The importance ranking of training indicators in the wakefulness dimension can be seen in Fig. 4.

Similarly, Fig. 5 shows the calculation results of the weight of the effectiveness dimension in the fifth step of the entropy method. The weights of each indicator should be analyzed based on the results. The maximum weighted value is 11.091% for *Test_MAPE*, with the MAP and RMSE of the test set and training set ranking in the top four.

In terms of the arousal emotion dimension, the model provides the most information about the four evaluation indicators in the test set. However, for the effectiveness dimension, the information content is highest for the *MAP* and *RMSE* indicators in both the test and training sets, with multiple indicators in the test set still ranking high.

Based on our experiments, we can conclude that the top four indicators mentioned above are more effective in evaluating the proposed model.

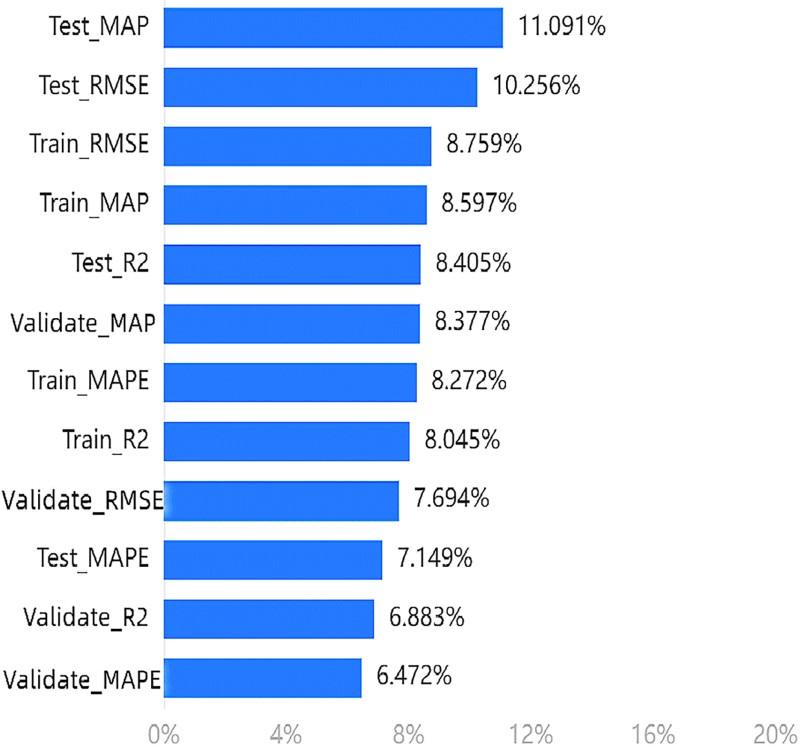

**Figure 5  Importance ranking of training indicators in the effectiveness dimension.**

## Performance comparison

We have compared the evaluation indicators of our model with those of other models. *Zhang et al. (2022)* proposed a novel method named modularized composite attention network (MCAN) for continuous MER. They conducted experiments on DEAM and PMEmo datasets. The superior results proved the effectiveness of their work. *Shih, Sun & Lee (2019)* proposed a novel attention mechanism to select relevant time series and used its frequency domain information for multivariate forecasting. They applied the proposed model to several real-world tasks and achieved state-of-the-art performance in almost all cases. In this section, we evaluated the performance of these models in both the awakening dimension and the valence dimension.

The results for the awakening dimension are shown in Table 1. In this dimension, our model has demonstrated significant improvements in accuracy and MAE compared to the MCAN model, which has been reported as the best-performing model in other studies. Specifically, our model has shown a 47.92% increase in accuracy and a 93.90% decrease in MAE. Additionally, it has exhibited excellent performance in other accuracy evaluation metrics. Furthermore, after optimization with the sparrow search algorithm, all evaluation metrics of our model have shown a significant enhancement in accuracy.

In terms of valence, as shown in Table 2, our model has outperformed the MCAN model with outstanding results compared to previous studies. It has improved RMSE accuracy by 55.63% and MAE by 93.83%. Additionally, it has shown impressive results in

**Table 1 Results of regression evaluation indicators of various research models in the awakening dimension.**

| Model | Data split | $R^2$ | RMSE | MAE | MAPE |
|---|---|---|---|---|---|
| MCAN | Not split | NAN | $0.135 \pm 0.057$ | $0.901 \pm 0.103$ | NAN |
| BCRSN | Not split | NAN | $0.132 \pm 0.059$ | $0.997 \pm 0.057$ | NAN |
| TNN+SVR | Not split | NAN | $0.145 \pm 0.022$ | $1.002 \pm 0.104$ | NAN |
| 3*TPA-BiLSTM | Training | 0.4973 | 0.1356 | 0.1433 | 12.41% |
| | Testing | 0.3771 | 0.1401 | 0.1595 | 13.90% |
| | Validation | 0.4702 | 0.1428 | 0.1508 | 12.83% |
| 3*ours | Training | 0.82705 | 0.073779 | 0.057976 | 10.65% |
| | Testing | 0.845 | 0.073959 | 0.058092 | 10.62% |
| | Validation | 0.81821 | 0.073934 | 0.057626 | 10.50% |

**Note:**
BCRSN, bidirectional convolutional recurrent sparse network.

**Table 2 Results of regression evaluation indicators of various research models in the valence dimension.**

| Model | Data split | $R^2$ | RMSE | MAE | MAPE |
|---|---|---|---|---|---|
| MCAN | Not split | NAN | $0.144 \pm 0.102$ | $0.892 \pm 0.030$ | NAN |
| BCRSN | Not split | NAN | $0.892 \pm 0.030$ | $0.892 \pm 0.030$ | NAN |
| TNN + SVR | Not split | NAN | $0.149 \pm 0.101$ | $0.889 \pm 0.059$ | NAN |
| 3*TPA-BiLSTM | Training | 0.4801 | 0.1374 | 0.1477 | 13.82% |
| | Testing | 0.2911 | 0.1564 | 0.1598 | 14.50% |
| | Validation | 0.4429 | 0.142 | 0.1501 | 13.83% |
| 3*ours | Training | 0.80279 | 0.071912 | 0.056649 | 10.61% |
| | Testing | 0.78649 | 0.072563 | 0.056937 | 10.49% |
| | Validation | 0.7975 | 0.069436 | 0.054371 | 9.91% |

**Note:**
BCRSN, bidirectional convolutional recurrent sparse network.

various accuracy assessment criteria. Moreover, after optimization with the sparrow search algorithm, all the evaluation metrics of our model have demonstrated significantly improved accuracy.

In summary, our proposed model demonstrates substantial performance improvements over existing methods such as MCAN and the model by *Shih, Sun & Lee (2019)* achieving a 47.92% increase in accuracy and a 93.90% decrease in MAE for the awakening dimension, and a 55.63% improvement in RMSE accuracy and a 93.83% decrease in MAE for the valence dimension. These enhancements underscore the model's effectiveness in capturing complex emotional nuances in music.

## Impact of lag order

The lag order has a significant impact on MER by influencing temporal dependencies within music and the emotional responses it triggers. In time series analysis and MER, the lag order refers to the number of previous time points used as predictors in a model. This significantly affects the accuracy of predictions and the understanding of the

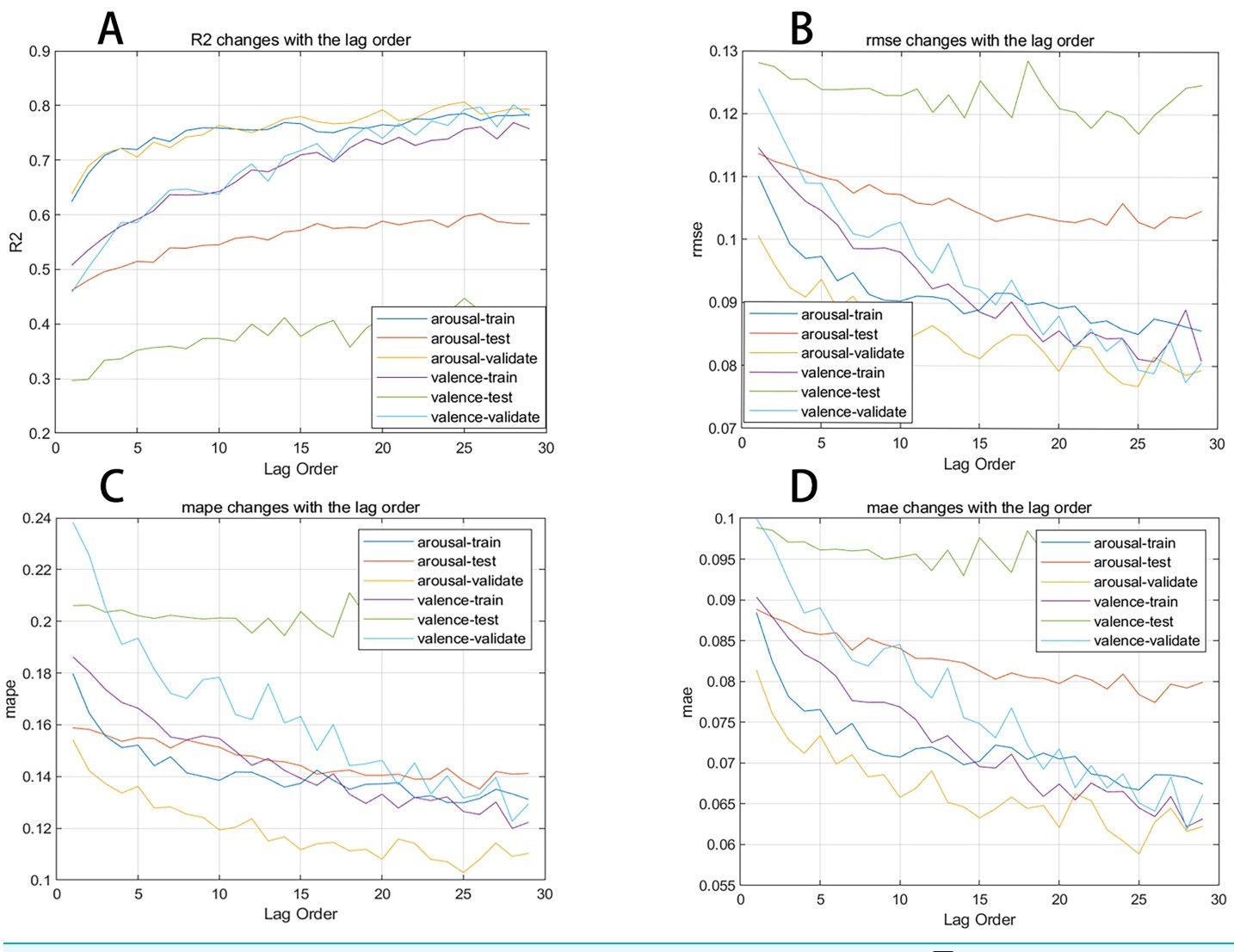

**Figure 6 (A–D) Evaluation metrics changing with lag order curve.**

data-generating process (DGP). The choice of lag order can dramatically impact the predictive performance of the MER model. An optimal lag order captures relevant past information while avoiding unnecessary data that could reduce prediction accuracy. This experiment examines how evaluation metrics vary across the training set, test set, and validation set with different lag orders to understand how the inclusion of lag orders affects the model's performance.

In this section, we plot the changes in evaluation metrics for the training, test, and validation sets during the prediction process of two emotional dimensions against different lag orders to present the results more clearly. Figure 6 shows the trend of the R2, MAE, RMSE, and MAPE evaluation metrics for the neural network as the lag order increases. We observe that the R2 for the arousal dimension in the training, test, and validation sets generally improves with the increase in lag order, while the MAE, RMSE, and MAPE overall decrease as the lag order increases.

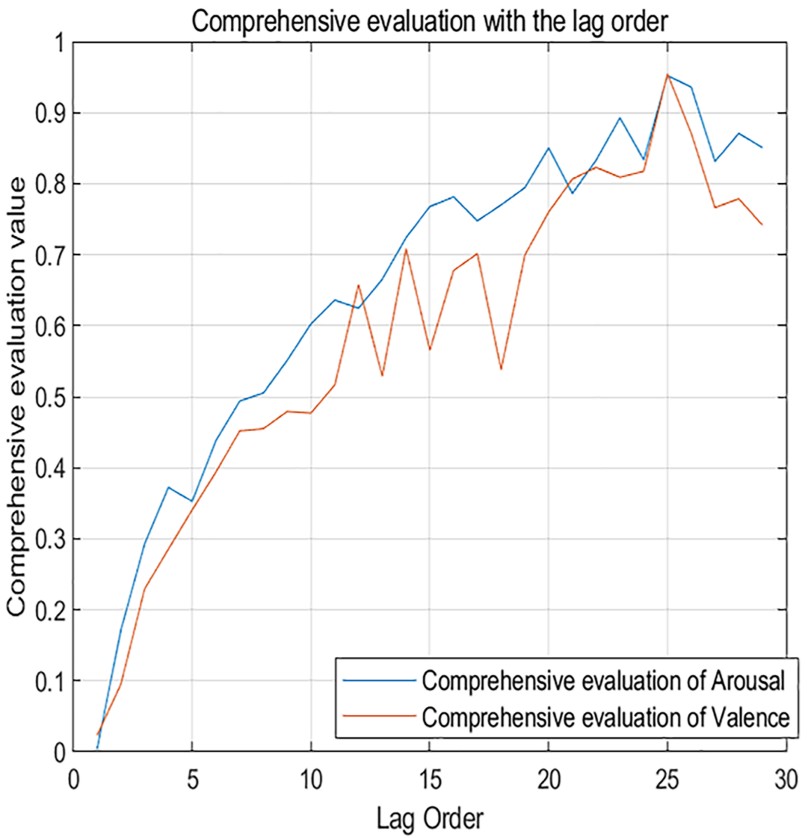

**Figure 7 Comprehensive evaluation metrics changing with lag order.**

**Table 3 Ablation study results for our model.**

| Model | RMSE | MAE | MAPE | R² |
|---|---|---|---|---|
| Full model | 0.043 | 0.033 | 0.055 | 93.92% |
| w/o CNN | 0.380 | 0.337 | 0.712 | 0.00% |
| w/o BiLSTM | 0.088 | 0.068 | 0.129 | 74.39% |
| w/o CNN and BiLSTM | 0.061 | 0.048 | 0.085 | 87.74% |

Figure 7 shows the changes in comprehensive evaluation scores from the 1st to the 29th order. The blue curve represents the arousal dimension, while the red curve represents the valence dimension. It's clear that as the lag order increases, the evaluation metrics for the neural network also increase. This confirms that dynamic music emotion prediction is a unique form of time series analysis. The network performs optimally when the lag order is 25, with a comprehensive evaluation score of 0.952 for the arousal dimension and 0.955 for the valence dimension.

## Ablation study

To further validate the contributions of each component to our proposed model, we conducted ablation studies on the PMEmo dataset for the arousal dimension. These studies

provide insights into how individual modules impact the overall performance of the model. As shown in Table 3, the full model demonstrates strong performance with an RMSE of 0.043, MAE of 0.033, MAPE of 0.055, and an $R^2$ of 93.92%. These metrics indicate the overall performance of the model when both the CNN and BiLSTM components are included.

The ablation study reveals the significance of each component to the model's performance. When the CNN module is removed ("w/o CNN"), there is a substantial increase in RMSE to 0.380, MAE to 0.337, and MAPE to 0.712, with a drop in $R^2$ to 0.00%. The CNN module is responsible for extracting local features from the input data. Its removal leads to a significant increase in RMSE, MAE, and MAPE, indicating that the model struggles to capture the essential patterns and features in the music data without it.

When the BiLSTM module is excluded ("w/o BiLSTM"), we observe a drop in $R^2$ to 74.39%, an increase in RMSE to 0.088, and an increase in MAE to 0.068, while MAPE increases to 0.129. This highlights the importance of the BiLSTM module in modeling the temporal dependencies and bidirectional context in the music data, which significantly enhances the model's ability to understand and predict emotional states.

The removal of both the CNN and BiLSTM modules leads to a significant decline in performance. The RMSE increases to 0.061, the MAE rises to 0.048, the MAPE grows to 0.085, and the $R^2$ value drops to 87.74%. These results demonstrate how both components enhance the model's overall performance by effectively capturing local features and temporal dependencies.

## CONCLUSION

This article presents an innovative CNN-BiLSTM model designed for music emotion regression. The model effectively integrates CNNs for feature extraction with BiLSTM networks for temporal modeling, further enhanced by a multi-head attention mechanism. This multi-scale processing enables the model to manage various levels of abstraction, while the inclusion of residual blocks helps to alleviate the vanishing gradient problem, resulting in more effective training and improved performance in predicting the emotional impact of music. Extensive testing on large datasets has resulted in an impressive R-squared value of 0.845, which is higher than the MCAN and bidirectional convolutional recurrent sparse network (BCRSN) models. These findings underscore the superior performance of our CNN-BiLSTM model in accurately predicting the emotional impact of music.

By accurately predicting the emotional impact of music, our model can serve as a powerful tool for therapists to tailor music-based interventions. For instance, it can help in selecting music that elicits specific emotional responses, thereby aiding in the treatment of conditions such as anxiety, depression, and PTSD. This personalized approach to music therapy can enhance therapeutic outcomes by providing targeted emotional support, ultimately contributing to improved mental health and well-being.

However, our work does have limitations. The model's performance is highly dependent on the quality of the annotated data, and the datasets used (DEAM and PMEmo) are primarily focused on Western music, which may restrict the model's applicability to other

musical cultures. To address these limitations, we plan to conduct cross-cultural studies, apply transfer learning techniques to adapt our model to datasets with limited annotations, and enrich our datasets with diverse cultural annotations in future work.

### Funding
This research received support from the Incubation Project of Xiamen University Tan Kah Kee College (Grant No. YY2022L03). The funders had no role in study design, data collection and analysis, decision to publish, or preparation of the manuscript.

### Grant Disclosures
The following grant information was disclosed by the authors:
Incubation Project of Xiamen University Tan Kah Kee College: YY2022L03.

### Competing Interests
The authors declare that they have no competing interests.

### Author Contributions
- Yi Qiu conceived and designed the experiments, analyzed the data, prepared figures and/or tables, authored or reviewed drafts of the article, financial support and provision of materials, and approved the final draft.
- Yu Lin conceived and designed the experiments, performed the experiments, analyzed the data, performed the computation work, prepared figures and/or tables, and approved the final draft.
- Yun Lin conceived and designed the experiments, analyzed the data, prepared figures and/or tables, authored or reviewed drafts of the article, and approved the final draft.

### Data Availability
The DEAM dataset is available at Kaggle: https://www.kaggle.com/datasets/imsparsh/deam-mediaeval-dataset-emotional-analysis-in-music.

The PMEmo dataset is available at GitHub: https://github.com/HuiZhangDB/PMEmo/tree/master.

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
