# Peer review of "Music audio emotion regression using the fusion of convolutional neural networks and bidirectional long short-term memory models"

_PeerJ Computer Science, doi:10.7717/peerj-cs.3086_

## Round 0.1 · original submission · Major Revisions

Reviewer 1 ·

Basic reporting

This paper introduces an innovative CNN-BiLSTM model for music emotion regression.

The manuscript is well-structured and adheres to academic standards, with clear language and relevant citations. However, the followings needs to be updated.

- In figure1, it would be better to add the proposed attention mechanism. Especially, how to apply attention mechanism to the proposed method is not well described.

- Additional descriptions of Eq(1) considering the difference between LSTM and the proposed LSTM is required.

- Is there any difference in Figure 3 comparing to basic LSTM structure? If so, please describe it. If not, I recommend to delete Figure 3.

Experimental design

The proposed model and methodology are well-defined, incorporating CNN, BiLSTM, and multi-head attention. While the study utilizes well-known datasets (DEAM, PMEmo), it lacks a discussion on potential dataset biases and their impact on generalization.

Furthermore, more details on hyperparameter selection and model training strategies would strengthen the reproducibility of the study.

Validity of the findings

The results indicate that the proposed CNN-BiLSTM model outperforms existing methods, with an impressive R² of 0.845. However, the statistical significance of these improvements is not thoroughly discussed, and additional ablation studies would help verify the contribution of each component. Moreover, the study should address potential limitations, such as its applicability to non-Western music genres.

Additional comments

This study presents a valuable contribution to the field of music emotion recognition and its application in psychological therapy. Nevertheless, a deeper exploration of real-world applications, such as case studies or user studies, would enhance its practical relevance. Finally, improving the clarity of some figures and providing supplementary materials, such as code and hyperparameter configurations, would be beneficial for future researchers.

·

Basic reporting

The present paper is quite well-written with professional and easy to follow English.
Check line 52 for the part ‘relaxation.be computationally’ and make necessary corrections.
In line 82, abbreviations to be defined beforehand for the term PTSD.

The work proposes a novel methodology combining CNN architecture with Bidirectional Long Short-Term Memory (BiLSTM) networks to analyze and predict the emotional impact of musical audio. For this the authors make use of two commonly used database of over 9000 music clips.
The authors have shown with the help of different widely used parameters such as RMSE, R2, MAPE that the model outperforms others while evaluating correctly the arousal and valence dimension in the annotated music emotion database.
Although the paper may be technically strong, though neural network architectures are not essentially my area of expertise, the title of the paper is quite misleading. The entire paper does not shed any light on how the accurate identification of emotion would be helping in psychology therapy.
The authors would benefit on elaborating how they propose this innovative regression model would help in psychology therapy in general. Otherwise drop that portion from the title and pitch the paper as a document of new technological advancement with a novel deep learning architecture, which I feel is the strong area of the paper.

Experimental design

The article content is well within the Aims and Scope of the journal. The two neural network architectures, CNN and BiLSTM have been combined here to form an overlap model, which has performed with higher classification accuracy in MER than the two architectures individually. All technical points related to the applied architecture have been rigorously reported here, like loss curves, accuracy etc., and detailed description of the tools applied have been provided with respective codes and algorithms. The novelty of the paper lies in the description and applicability of this overlap model, and the relevant technical details have been elaborately addressed and explained.

Validity of the findings

Emotion Classification with neural network has become a very popular research domain in recent times and numerous works have been reported so far. In that matter, it becomes hard to add any innovative approach to this field. But, the novelty of this work lies in the fact that here an CNN-BiLSTM overlap model has been proposed and applied, which successfully shows higher classification accuracy with the large dataset of 9000 clips.
But, as the title suggests, the findings do not discuss or explore any points within the domain of psychology therapy. My main area of interest for the work, was the application of the proposed neural network overlap model of CNN-BiLSTM for assessing therapeutic effects of music, which has not been addressed here. All the points covered in the ‘Conclusion’ section discuss the technicality of the model and not its applicability to music intervention therapy. This deviates from the focal idea with which the work started, and the title also becomes somewhat misleading in that case. Leveraging MER in the discipline of scientific music therapy is indeed a potent area of exploration and would have added ample novelty to the paper apart from being technically sound one.

Additional comments

Authors should either consider changing the title according to the points reported in the paper, or they should consider incorporating valid studies and discussions regarding the application of the CNN-BiLSTM overlap model for psychology therapy, which also is quite a vast field of research and comes with its own challenges and complications. How well the proposed model fits into this domain requires rigorous studies. Either such points should be reported in the paper, and if the authors decide to focus only on the technical details and applied algorithms, then the title should be accordingly changed.

---

## Round 0.2 · accepted · Accept

The reviewers appreciated the changes made by the authors to the manuscript and therefore I can recommend this article for acceptance.

Reviewer 1 ·

Basic reporting

The concerns I raised regarding the attention mechanism and its placement in the model were thoroughly addressed. The updated Figure 1 now clearly shows the attention layer, and the corresponding section in the methodology provides a detailed explanation of how it works and how it contributes to refining feature representations before the final prediction layer. This significantly improves the interpretability of the model structure.

In addition, the authors provided an extended explanation of Equation (1) and clarified how their BiLSTM implementation differs from the standard LSTM. I found this clarification helpful, especially in the context of how forward and backward hidden states are used together.

Regarding Figure 3, I appreciate that the authors didn’t just keep it for formality but took the time to redraw and contextualize it properly. The revised diagram makes the bidirectional structure explicit, which was previously ambiguous. These updates collectively improve the clarity and completeness of the manuscript.

Experimental design

I had previously pointed out the lack of discussion around dataset bias and training strategies. The revised version now explicitly acknowledges potential cultural and demographic biases in the DEAM and PMEmo datasets. I find this addition important, as it reflects an awareness of generalization challenges—especially relevant for music emotion recognition tasks across diverse populations.

Furthermore, the authors have now provided a clear description of the training setup, including how hyperparameters were selected and how data was preprocessed and augmented. These details make the study much more reproducible and trustworthy.

Validity of the findings

The performance results were already impressive, but my concern was the lack of statistical context. The authors have now added comparative statistics showing significant improvements over previous models in terms of R² and MAE, and they quantify these differences clearly.

More importantly, the newly added ablation study is a strong contribution. It isolates the effects of each module (CNN, BiLSTM) and shows how their removal affects performance. This validates the architectural choices made in the proposed model and gives readers a deeper understanding of component-wise contributions.

I also appreciate the honest discussion in the conclusion about the model’s limited applicability to non-Western music and the commitment to explore this in future work. It’s important to see limitations acknowledged, and I think this addition strengthens the paper.

Additional comments

The authors have outlined plans to explore real-world applications of their model, including potential user studies, which is great to see. They’ve also improved the readability of figures, and I can confirm that the visuals (particularly Figures 1 and 3) are now clearer and more informative.

Finally, the decision to release the code and configurations via GitHub is a welcome move. It aligns well with open research practices and will be helpful to others working in this space.

·

Basic reporting

The manuscript is well-written with easy understanding, and aligns with the Aims and Scopes of the Journal. All points mentioned in the reviews have been addressed properly in detail. The title has also been modified and is now in sync with the work.

Experimental design

-

Validity of the findings

-